Eating for numbing: a community-based study of trauma exposure, emotion dysregulation, dissociation, body dissatisfaction and eating disorder symptoms

http://orcid.org/0000-0001-5565-2506 Lev-ari Lilac 1 2 ldlevari@gmail.com
Zohar Ada H. 1 2
Bachner-Melman Rachel 1 3
1 Clinical Psychology, Ruppin Academic Center , Emek Hefer , Israel
2 The Lior Tsfaty Center for Suicide and Mental Pain Studies, Ruppin Academic Center , Emek Hefer , Israel
3 School of Social Work, Hebrew University of Jerusalem , Jerusalem , Israel
Notebaert Lies
Electronic publication date: 2021 Aug 5
Publication date: 2021
Volume: 9
Electronic Location ID: e11899
Received 2021 Mar 3; Accepted 2021 Jul 13
Copyright: © 2021 Lev-ari et al.
Copyright year: 2021
Copyright holder: Lev-ari et al.
License: This is an open access article distributed under the terms of the Creative Commons Attribution License, which permits unrestricted use, distribution, reproduction and adaptation in any medium and for any purpose provided that it is properly attributed. For attribution, the original author(s), title, publication source (PeerJ) and either DOI or URL of the article must be cited.
License URL: https://creativecommons.org/licenses/by/4.0/

Keywords: Eating disorder symptoms, Trauma, Emotional dysregulation, Body dissatisfaction, Dissociation

Funding: The authors received no funding for this work.

==============================
Objective

The current study tests the relationship between eating disorder (ED) symptoms and trauma exposure. The mechanisms via which trauma is related to ED symptoms have not been sufficiently examined. This study examines the complex role of dissociation and emotional dysregulation in the context of trauma, BMI, ED symptoms and body dissatisfaction (BD). We hypothesized that dissociation and emotional dysregulation would mediate the relationship between trauma exposure and ED symptoms/BD. We further hypothesized that BMI would play a moderating role in this association.

Method

A community sample of 229 (16.2% male) participants, with a mean age of 29.08 ± 10.68 reported online on traumatic events (Life Events Checklist), dissociation (Dissociative Experiences Scale-II), emotional dysregulation (Difficulties in Emotional Regulation Scale), ED symptoms (Eating Disorders Examination-Questionnaire) and BD (Figure Rating Scale).

Results

Participants reported experiencing a mean of 2.87 ± 2.27 traumatic events, with a relatively high percentage (~86%) reporting at least one. The most commonly reported traumatic events were transportation accidents and physical assault. Although frequency of traumatic events did not directly predict ED symptoms, BMI, dissociation, emotional dysregulation and BD did. An SEM model showed that traumatic events predicted ED symptoms indirectly through dissociation, emotional dysregulation and BMI. Dissociation and emotional dysregulation predicted ED symptoms directly. BMI also moderated the association between traumatic events and both ED symptoms and BD.

Conclusions

Therapists treating patients with high BMI or obesity should be aware of these relationships and investigate the possibility that trauma and/or PTSD may underlie the presenting disordered eating or eating disorder.

Introduction

Disordered eating (DE) behaviors and attitudes make up part of the eating disorder continuum. Attitudes include obsessing about food and calories, becoming angry when hungry, being unable to select what to eat, seeking food to compensate for psychological problems, eating until feeling sick, and presenting unreal myths and beliefs about eating and weight (American Dietetic Association, 2006). A more parsimonious definition is suggested by the work of Hazzard et al. (2021), which is that any deviation from eating according to hunger and satiation cues, i.e. intuitive eating, is disordered. In practice, disordered eating is measured by assessing symptoms of eating disorders, behavioral, cognitive, and emotional (Alvarenga, Scagliusi & Philippi, 2010), which do not reach the threshold of an eating disorder diagnosis. The current study examines exposure to traumatic events, the absence of effective regulation of negative emotion and the presence of dissociation as potential predictors of DE. These may serve to numb the unprocessed distress arising from the experience of the trauma.

Whereas emotional dysregulation and dissociation may sometimes play a positive role in dealing with stressful events, they have the potential to become maladaptive stress response mechanisms that impair the individual’s ability to function effectively. If they persist and become permanent, emotional dysregulation and dissociation can hinder a person’s way of dealing with everyday life stressors. A recent systematic review and meta-analysis of the literature showed that adults with Post Traumatic Stress Disorder (PTSD) who survived adverse events in childhood commonly experience emotional dysregulation and dissociation (Boyd et al., 2020). These play an important role in functional impairment among women with a history of childhood abuse beyond PTSD and seem to impair cognition, mobility, daily functioning, self-care, community involvement and interpersonal connection at least as much as PTSD symptoms (Cloitre et al., 2005).

Whereas the role of trauma exposure has previously been implicated in eating disorder symptomatology, the mechanisms that may underlie this relationship remain uncertain. One such mechanism passes through dissociation, a of state of consciousness during which events and experiences usually integrated in consciousness are divided off and separated from one other. This allows more than a single stream of consciousness, or multi-tasking. In her seminal book “Trauma and Recovery”, Herman (2015) explains the process of dissociation in the context of trauma. After experiencing traumatic events, especially persistent trauma, individuals may adopt dissociation as a routine way of dealing with everyday stressors. What begins as an adaptive response can eventually impair the ability to focus on and respond appropriately to internal and external circumstances.

Severe dissociative disorders are closely associated with exposure to trauma. Somer (2004) found that the higher the intensity of the reported trauma, the stronger the dissociative effects. He concluded that if the traumatic experiences are particularly prolonged and violent, the person has little hope for relief, and his/her suffering is unrecognized, dissociation can maintain the illusion of self-oneness and self-continuity, allowing the person to avoid experiencing unbearable realities (Somer, 2004). When parts of the self are disconnected and disowned, the self develops further around the dissociative axis, fostering the creation of several narratives that allow for the coexistence of the different parts of self. A dissociative inner organization may be life-saving but also enables a pseudo-existence, a zone of half-life (Seligman, 2017). Dissociative defenses have two opposite effects. On the one hand, they help the person move away from the horror of the traumatic experience, but on the other they disrupt processing and recovery after the injury (Somer, 2004). If the event is experienced as shameful and secretive, dissociation does not allow the trauma to be processed (Schwarzberg & Somer, 2004).

Gruhn & Compas (2020) provide evidence that maltreatment early in life can trigger processes that lead to emotion dysregulation. They point out that the emotional response patterns of avoidance, suppression, and emotional coping can be helpful in surviving childhood maltreatment. However, if generalized, they can become part of the diathesis for future psychopathology. Alternations in neural development (Tupler & De Bellis, 2006) and impairments in biological stress response may occur (Tarullo & Gunnar, 2006), and genetic diathesis might render some individuals even more vulnerable than others to emotional dysregulation (Halldorsdottir et al., 2017).

There is some support for a specific relationship between exposure to trauma and eating pathology. In their review, Trottier & MacDonald (2017) present evidence that exposure to childhood trauma and severe adverse events are associated with bulimia nervosa (BN), binge eating disorder (BED) and anorexia nervosa (AN), particularly the binge-purge subtype (ANBP). Furthermore, they present some evidence from longitudinal studies for a prospective association between exposure to trauma as an adult and eating pathology. In adult women, sexual assault was found to be related to binging and purging behaviors; the more violent the assault, the more extreme was the eating pathology. Similar findings have been reported for disordered eating in non-clinical samples. Hasselle et al. (2017) found that university students who reported more childhood victimization also reported more disordered eating and higher levels of emotion dysregulation. Other studies focus on PTSD or other psychological symptoms rather than the exposure to trauma. Mitchell et al. (2016), for instance, reported that the relationship between PTSD symptoms and disordered eating, food addiction and overweight in elderly male veterans was mediated by the suppression of expressions of fear, hostility, guilt and sadness, underscoring a specific link between trauma, problems in emotional expression and disordered eating.

Palmisano et al. (2018) conducted a case control study of patients with obesity and no ED, patients with BED, and controls with average body mass index (BMI; kg/m2) They found that the participants with obesity reported higher levels of dissociation, and within the obese group, those with BED reported more childhood trauma than those without. Longo et al. (2019) found that in an ED treatment center, patients with ANBP reported more traumatic life events, especially sexual trauma, prior to the onset of the ED than patients with restrictive AN.

A relationship between exposure to trauma, psychological distress and high BMI has been demonstrated in several contexts. Gu et al. (2013) conducted a national survey of law enforcement officers over six years and found that those who reported higher levels of psychological distress, likely related to exposure to traumatic events, were also heavier; when controlling for all potential confounders, this was true for women but not men. Hampson et al. (2016) followed a community sample of children in Hawaii from childhood to adulthood and found that teen and adult trauma exposure was related to higher BMI and greater risk for obesity in adult women. In a longitudinal study of women nurses, Kubzansky et al. (2014) found that women with at least four PTSD symptoms gained more weight than other women and were at increased risk for obesity. Schrepf, Markon & Lutgendorf (2014), studied adults who retrospectively reported on childhood trauma. For both men and women there were associations between reported childhood trauma, emotional eating, current obesity, and risk factors for cardiovascular disease. Risk for obesity therefore appears to be linked to trauma exposure, PTSD and psychological distress.

Moulton et al. (2015) and Pugh, Waller & Esposito (2018) found that dissociation mediated the association between exposure to trauma during childhood and eating psychopathology in adulthood. These studies showed an indirect relationship between exposure to trauma and disordered eating via dissociation. Other studies, however, have reported mixed findings on dissociation as a possible mediator between exposure to trauma and disordered eating (Gerke, Mazzeo & Kliewer, 2006; Kent, Waller & Dagnan, 1999). Fox & Power (2009), for example, surmised that dissociation might, in fact, be a form of emotion regulation.

The current study was conducted in Israel. Israel has a compulsory 2/3-year mandatory military service with enlistment at 18 years of age, and the country experiences frequent military conflicts (Sasson-Levy, 2007). Thus, the prevalence of traumatic experiences might be higher than in other westernized countries (Hoffman, Diamond & Lipsitz, 2011). This study examines relationships between exposure to trauma, emotion dysregulation, dissociation, disordered eating (operationalized in terms of ED symptoms) and BD. Previous studies have linked trauma to dissociation and dissociation to emotional dysregulation. Dissociation and emotional dysregulation have also been linked to ED symptoms and BD. Therefore, we aimed to build a model that shows the influence of dissociation and emotional dysregulation on the relationship between trauma and ED symptoms/BD. Our hypotheses were:Frequency of exposure to traumatic events would be significantly correlated with dissociation, emotional dysregulation, BD, ED symptoms and BMI.

BMI, frequency of exposure to traumatic events, dissociation, emotional dysregulation, and BD would predict ED symptoms (while controlling for demographic variables).

Dissociation and emotional dysregulation would mediate the association between exposure to traumatic events and the other variables—BD, BMI and ED symptoms.

BMI would influence the association between frequency of exposure to traumatic events and BD/ED symptoms.

Methods

Participants

A total of 233 (40; 17.2% males) participants between 20 and 72 years of age (M = 29.81, SD = 11.96) registered online to participate in the study. Of these, 80 were community volunteers recruited via social networks and 153 were students enrolled in a BA level introductory psychology course who received credit for their participation. Almost three-quarters (74.7%) were single, 50 (21.5%) were married and nine (3.8%) were divorced or had “other” status. They had 0–6 children (M = 0.48, SD = 1.12) and a mean of 13.22 years of schooling (SD = 2.59). Their BMI (kg/m2) ranged between 16.36 and 49.54 (M = 23.16, SD = 4.64), with most (83.2%) falling in the mid-BMI range (18.5 < BMI < 25). A minority (8.6%) had low BMIs (<18.5) and 8.2% had high BMIs (>25).

Measures

Traumatic experiences

Traumatic experiences were assessed using the Life Events Checklist (LEC; Gray et al., 2004). The LEC contains 17 items dealing with potentially traumatic events (e.g. fire/explosion, combat/exposure to war zone) and screens for post-traumatic experiences. Each item is rated qualitatively between a–e (a = happened to me, b = I witnessed it, c = heard about it, d = unsure, e = inappropriate), and participants who indicated that an event had happened to them (a) were considered to have experienced this trauma. Frequency of traumatic experiences was operationalized as the number of events participants indicated they had experienced. Gray et al. (2004) assessed 108 undergraduate psychology students and found that the LEC had good construct validity. We used the Hebrew translation of this questionnaire (Goren, 2003).

Dissociation

Dissociation was assessed using the Dissociative Experiences Scale-II (DES-II; Carlson & Putnam, 2001). The DES-II contains 28 items that describe different types of dissociative experiences including amnesia (e.g. “Some people have the experience of driving or riding in a car or bus or subway and suddenly realizing that they don’t remember what has happened during all or part of the trip”), depersonalization and de-realization (e.g. “Some people sometimes have the experience of feeling as though they are standing next to themselves or watching themselves do something and they actually see themselves as if they were looking at another person”), and absorption (“Some people find that when they are watching television or a movie they become so absorbed in the story that they are unaware of other events happening around them.”). Respondents are asked what percent of the time they experience each kind of dissociative experience on a categorical scale from 0% of the time i.e. never to 100% of the time i.e. always. Level of dissociation is calculated by averaging questionnaire items for each participant, with higher scores describing higher levels of dissociation. Somer, Dolgin & Saadon (2001) assessed 340 consecutive admissions to an Israeli outpatient clinic and 290 non-clinical participants and found good test-retest reliability using the Hebrew translation (0.87). In the current study the DES-II had high internal reliability (Cronbach’s alpha = 0.90).

Emotional regulation

Difficulties in emotional regulation were assessed using the Difficulties in Emotional Regulation Scale (DERS; Gratz & Roemer, 2004). The DERS contains 36 statements about various aspects of emotional regulation, such as “When I’m upset, I lose control over my behavior”. Respondents rate their level of agreement with each statement on a scale from 1 (almost never) to 5 (almost always). The degree of difficulty in emotional regulation is calculated by averaging the items in the questionnaire, with higher scores reflecting greater difficulty regulating emotions. Reuveni et al. (2016) assessed 648 female university students and found good internal reliability (0.92). We used the Hebrew version of this questionnaire (Segal & Golan, 2016), which showed good internal reliability (Cronbach’s alpha = 0.90).

Eating disorder symptoms

ED Symptoms were assessed using the Eating Disorders Examination-Questionnaire (EDE-Q; Hilbert et al., 2007). The EDE-Q contains 28 items assessing the core symptoms of eating disorders and a range of associated pathology, including the frequency of undereating, overeating, dysregulation and compensation. The EDE-Q has four subscales, each containing five to eight items: (1) Dietary Restraint (DR; “Have you been deliberately trying to limit the amount of food you eat to influence your shape or weight (whether or not you have succeeded”)?); (2) Eating Concern (EC; “Over the past 28 days, how concerned have you been about other people seeing you eat?”); (3) Weight Concern (WC; “Have you had a definite fear that you might gain weight?”); and (4) Shape Concern (SC; “Have you had a definite desire to have a totally flat stomach?”). Twenty-two items are rated on a seven-point Likert-type scale ranging from 0 (never) to 6 (always). Higher scores reflect greater symptom severity. The remaining six items concerning the frequency of weight, shape and use of purging techniques during the past 28 days require numerical responses (from 0 to 28). These six items are used for diagnostic purposes and are generally excluded from factor analysis. The cut-off score of four (in subscales and in the global score) is suggested as being indicative of clinical EDs for both men and women (Luce, Crowther & Pole, 2008). In this study, disordered eating or “ED symptoms” refers to the global EDE-Q score (average of all EDE-Q subscales). Zohar, Lev-Ari & Bachner-Melman (2017) assessed 292 community volunteers and found sound psychometric properties for the Hebrew translation but recommended combining WC and SC into one subscale (Zohar, Lev-Ari & Bachner-Melman, 2017). In the current study the internal reliability was acceptable (Cronbach’s alpha > 0.78).

Body dissatisfaction

Body dissatisfaction was assessed using the Figure Rating Scale (FRS; Stunkard, Sorenson & Schlusinger, 1983). The FRS presents silhouettes of increasing size, along a scale numbered from one through nine. Respondents are asked to choose: (1) the figure that best represents their own body; (2) the figure that best represents their ideal body. BD is calculated by subtracting the numbered silhouette of the ideal body from the numbered silhouette of the current body. The FRS is one of the most widely used instruments in body image research (e.g., Bjerggaard et al., 2015; Easton, Stephens & Sicilia, 2017; Gardner & Brown, 2010), and has good psychometric qualities (Cohn et al., 1987). Thompson & Altabe (1991) observed high test-retest reliability for the FRS in a sample of 58 females (r = 0.70–0.90).

Procedure

The Institutional Internal Review Board approved the study and granted ethical approval to carry out the study within its facilities (Ethical Application Ref: 35 Tashaz). Volunteers interested in participating in the study were sent a computerized link to the questionnaires. On the first screen of the online self-report, they received a full explanation about the study and were asked to provide informed consent. They reported on demographic information, height and weight, then completed the LEC, DES, EDE-Q and the DERS. After questionnaire completion, contact details of the researchers were provided and participants were encouraged to reach out to them with any questions, comments or difficulties.

Statistical analysis

Missing data: Participants were prompted to respond to unanswered questions. All participants answered the entire questionnaire with the exception of one participant for whom one response was missing. The mean of this scale was assessed without the missing response. Two participants did not answer the questions about education, three did not offer their age and 21 did not report how many children they had. Demographic data is reported without these missing responses.

Descriptive statistics were used to quantify frequency of trauma and Pearson correlations to quantify the associations between variables. Hierarchical Regression Analysis was used to predict ED symptoms using the independent variables. Structural equation model (SEM) was employed to assess the mediating effect of emotional dysregulation and dissociation on the association between trauma and both ED symptoms and BD. Mediation analyses were conducted to assess the effects of BMI on the association between trauma and BD/ED symptoms. Results did not change significantly when subscales versus global scores were used, therefore only global scores were reported. SPSS 23.0 and AMOS 23.0 were used for statistical analyses.

Results

Hypothesis 1: Frequency of exposure to traumatic events would be significantly correlated with dissociation, emotional dysregulation, BD, ED symptoms and BMI.

Descriptive statistics

A high percentage of participants (~86%) reported having experienced at least one traumatic event. Table 1 presents the percentage of participants who reported having experienced each type of traumatic event. The most reported traumatic events were transportation accidents and physical assault. Fully 41% reported a traumatic experience in battle. About 10% of participants reported having been sexually assaulted and almost half (~46%) had had an unwanted sexual experience. Since many participants reported having experienced more than one traumatic event, Fig. 1 presents the cumulative frequency of traumatic events experienced.

Table 1 Percent of participants who reported having experienced traumatic events (N = 233).

Traumatic event	Percent	
Transport accident	67.63	
Physical assault	52.59	
Sudden death of someone close to you	50.32	
Unwanted sexual experience	45.63	
Battle	41.20	
Life-threatening illness	14.78	
Fire	14.78	
Work accident	14.78	
Natural disaster	13.66	
Exposure to toxic material	6.39	
Sexual assault	10.43	
Assault with weapons	7.87	
Human agony	5.91	
Violent death of someone close to you	1.30	
You hurt or killed someone	1.30	
Captivity	0.87	
Other	56.38	

Figure 1 Frequency of traumatic events (N = 233).

Participants reported experiencing between 0–13 traumatic events (Mean = 2.87, S.D. = 2.27) (see Fig. 1). Nearly half (48.5%) reported experiencing up to two traumatic events, and 3% reported experiencing eight or more.

One-sided Pearson correlations were calculated to assess the associations between variables (see Table 2). The frequency of exposure to traumatic events was positively correlated with dissociation, emotional dysregulation and BMI, but not with ED symptoms or BD. Dissociation was positively correlated with emotional dysregulation and ED symptoms but not with BMI or BD. Emotional dysregulation was positively correlated with BD and ED symptoms but not with BMI.

Table 2 Pearson correlations between frequency of traumatic events and dissociation, emotional dysregulation, BD, ED symptoms and BMI (N = 233).

	LEC	DES-II	DERS	FRS	EDE-Q	BMI	
Frequency of traumatic events (LEC)*		0.17**	0.15**	−0.08	0.09	0.14**	
Dissociation (DES-II)			0.39***	0.10	0.28***	−0.02	
Emotional dysregulation (DERS)				0.22***	0.33***	−0.04	
Body dissatisfaction (FRS)					0.12***	0.23***	
ED symptoms (EDE-Q)						0.33***	
Mean
(SD)	2.87 (2.27)	2.78 (1.36)	2.20 (0.54)	1.00 (1.43)	1.69 (1.42)	23.16 (4.64)	
Notes:

* p < 0.05.

** p < 0.01.

*** p < 0.001.

LEC, life events checklist; DES-II, dissociative experiences scale-II (total score); DERS, difficulties in emotional regulation scale (total score); EDE-Q, eating disorders examination-questionnaire (global score); FRS, figure rating scale.

Hypothesis 2: BMI, frequency of exposure to traumatic events, dissociation, emotional dysregulation, and BD would predict ED symptoms (while controlling for demographic variables).

To explore the effects of BMI on the various indices, we divided our sample into three groups according to their BMI: High (>25; n = 20); intermediate (18.5–25; n = 193); and low (<18.5; n = 19) and assessed between-group differences on all variables (frequency of exposure to traumatic events, dissociation, emotional dysregulation, body dissatisfaction and ED symptoms). MANOVA analysis revealed overall significant differences between all three groups (F(10,450) = 3.71, p < 0.001), but only ED symptoms (F(2,228) = 10.81, p < 0.001) and BD (F(2,228) = 8.33, p < 0.001) differed significantly between the groups, echoing the findings of the Pearson correlations.

To examine the hypothesis that BMI, frequency of exposure to traumatic events, dissociation, emotional dysregulation, and body dissatisfaction would predict ED symptoms (while controlling for demographic variables), a hierarchical regression analysis was conducted. Demographic variables (age, sex and education) were introduced in the first step in order to control for them, making a case that they were not the reason for further associations between variables. BMI was added in the second step as previous studies have found associations between both BMI and trauma and between BMI and ED symptoms. Adding BMI in this step allowed for assessing its added influence on explained variance and allowing to control for it when assessing the association between trauma and ED symptoms. Frequency of exposure to traumatic events and dissociation were added in the third step to assess their unique contribution to the explained variance as were emotional dysregulation and BD in the last step. Results are presented in Table 3.

Table 3 Hierarchical regression analysis predicting ED symptoms (global score) from BMI, frequency of traumatic events, dissociation, emotional dysregulation, and body dissatisfaction, while controlling for demographic variables (N = 233).

	Standardized β	T	Adj. R2	ΔR2	F(df)	p	
Step 1			0.006		1.49(3,223)	NS	
Sex	0.12	1.73					
Education	−0.08	−0.96					
Age	0.12	1.40					
Step 2			0.12	0.11***	8.38***(4,222)	0.00	
Sex	0.16	2.34*				0.02	
Education	−0.05	−0.63				0.53	
Age	−0.02	−0.22				0.83	
BMI	0.36	5.34***				0.00	
Step 3			0.17	0.06***	8.61***(6,220)	0.00	
Sex	0.15	2.26*				0.03	
Education	−0.01	−0.07				0.95	
Age	−0.04	−0.05				0.96	
BMI	0.35	5.36***				0.00	
LEC	0.01	0.21				0.84	
DES-II	0.25	3.89***				0.00	
Step 4			0.23	0.06***	10.61***(7,219)	0.00	
Sex	0.17	2.58**				0.01	
Education	0.03	0.43				0.67	
Age	0.05	0.62				0.54	
BMI	0.34	5.41***				0.00	
LEC	−0.01	−0.20				0.84	
DES-II	0.16	2.49**				0.01	
DERS	0.28	4.30***				0.00	
Step 5			0.31	0.08***	13.78***(8,218)	0.00	
Sex	0.07	1.16				0.25	
Education	−0.01	−0.10				0.93	
Age	0.08	1.09				0.28	
BMI	0.24	3.83***				0.00	
LEC	0.01	0.23				0.82	
DES-II	0.16	2.60**				0.01	
DERS	0.21	3.30***				0.001	
FRS	0.32	5.21***				0.00	
Notes:

* p < 0.05.

** p < 0.01.

*** p < 0.001.

Sex (1 = male; 2 = female); BMI, body mass index; LEC, life events checklist; DES-II, dissociative experiences scale-II (total score); DERS, difficulties in emotional regulation scale (total score); FRS, figure rating scale.

As can be seen from Table 3, BMI, dissociation, emotional dysregulation and BD were all uniquely associated with ED symptoms. BMI, dissociation, emotional dysregulation and BD were associated with ED symptoms. The frequency of exposure to traumatic events did not predict ED symptoms.

Hypothesis 3: Dissociation and emotional dysregulation would mediate the association between exposure to traumatic events and the other variables—BD, BMI and ED symptoms.

A central aim of this study was to build a comprehensive model depicting the relationships between exposure to traumatic events, dissociation, emotional dysregulation, BD, BMI and ED symptoms. A structural equation model (SEM) was designed, following the recommendation of Hayes (2009). Since dissociation, emotional dysregulation and body dissatisfaction were hypothesized to be mediating variables, they were entered as such in the analysis. Since BMI, BD and ED symptoms are known to be highly intercorrelated, they were entered together as correlated mediators in the model. In order to take into account, the prediction of BD on emotional dysregulation, the path that was entered into the analysis in this order. As a combined rule for the acceptance of our model, we chose the following acknowledged values: normed fit index (NFI) > 0.90 (Bentler & Bonett, 1980) and root mean square error of approximation (RMSEA) < 0.08 (Browne & Cudeck, 1993; see Fig. 2). The Chi Square goodness-of-fit index presented an excellent fit for the data, χ(5)2 = 5.66, p = 0.34; NFI = 0.97; CFI = 0.99; RMSEA = 0.02; standardized root means square residual (RMR) = 0.03.

Figure 2 SEM model depicting the relationships between frequency of traumatic events, dissociation, emotional dysregulation, BD, BMI and ED symptoms.

Note: *p < 0.05; **p < 0.01; ***p < 0.001. Frequency of traumatic events = LEC; Emotional dysregulation = DERS (total score); Dissociation = DES-II (total score); BMI, body mass index; Body dissatisfaction = FRS; ED symptoms = EDE-Q (global score).

The frequency of exposure to traumatic events positively predicted dissociation, emotional dysregulation and BMI. BMI and BD were positively associated with one another, and BD positively predicted emotional dysregulation. BMI, BD and ED symptoms were all positively associated. Exposure to traumatic events did not directly predict ED symptoms, but predicted them indirectly through dissociation, emotional dysregulation and BMI. Dissociation and emotional dysregulation directly predicted ED symptoms.

Hypothesis 4: BMI would influence the association between frequency of exposure to traumatic events and BD / ED symptoms.

After noting the model of associations between frequency of traumatic events and ED symptoms and BD, we further hypothesized that BMI would moderate the relationship between frequency of traumatic events and BD and ED symptoms.

Two moderation analyses were conducted. A hierarchical regression model was built to conduct a moderation analysis between frequency of traumatic events, BMI and BD. Frequency of traumatic events and BMI underwent centering procedures. In the first stage, frequency of traumatic events and BMI were entered as independent variables and BD (FRS) as the predictor. The model was statistically significant (F(2,229) = 8.22, p < 0.001) and explained 5.9% of the variance. The addition of the interaction between frequency of traumatic events and BMI to the second model was statistically significant (F(3,228) = 20.77, p < 0.001), adding an extra 15.57% to the explained variance. Analyses using the 5000-bootstrap method (Hayes, 2017) confirmed that BMI moderated the association between frequency of traumatic events and BD (See Fig. 3A).

Figure 3 Moderation analysis: BMI moderates the association between frequency of traumatic events and (A) BD and (B) ED symptoms (global score).

A follow-up analysis was conducted, aimed at estimating the conditional effects of the predictor variable (trauma frequency) on the outcome variable (BD) at low and high values of the moderator (BMI). The model for low BMI was not significant. For high BMI, the model was statistically significant (F(1,113) = 8.55, p = 0.004), explaining 6.2% of the variance. Trauma frequency had a negative (β = −0.27) and statistically significant effect on BD (t = −2.93, p = 0.004).

A hierarchical regression model was then built to conduct a moderation analysis between frequency of traumatic events, BMI and ED symptoms. Frequency of traumatic events and BMI underwent centering procedures. In the first stage, frequency of traumatic events and BMI were entered as independent variables and ED symptoms (EDE-Q) as the predictor. The model was found to be statistically significant (F(2,229) = 13.94, p < 0.001), explaining 10.0% of the variance. The addition of the interaction between frequency of traumatic events and BMI to the second model was statistically significant (F(3,228) = 11.80, p < 0.001), adding an extra 3.44% to the explained variance. Analyses using the 5000-bootstrap method (Hayes, 2017) confirmed that BMI moderated the association between frequency of traumatic events and ED symptoms (See Fig. 3B).

A follow-up analysis aimed at estimating the conditional effects of the predictor variable (trauma frequency) on the outcome variable (EDE-Q global score) at low and high values of the moderator (BMI) was conducted. Neither model (for low or high BMI) was statistically significant.

As can be seen in Figs. 3A and 3B: For people with low BMI, those with less traumatic events, also had the least ED symptoms. People with low BMI and many traumatic events had more ED symptoms, but these were still less than people with high BMI. People with high BMI had higher ED symptoms, regardless of the frequency of their traumatic events in the past.

Discussion

The main purpose of this study was to examine the association between the frequency of traumatic events and both BD and disordered eating, as well as the influence of emotional dysregulation, dissociation and BMI on this association.

Participants in this community study reported a wide range of traumatic experiences, with around half the sample reporting that they experienced up to two traumatic events in their lifetime. The most common events reported were transport accidents, physical assault, and the sudden death of a close person. Almost half the participants reported having had unwanted sexual experiences, and 10% a sexual assault. Over 40% reported having experienced trauma in battle. This surprisingly large proportion can be explained by the fact that two or three years of military service is compulsory for the participants, who were young Israeli adults, many of whom had experienced combat. Beyond the idea of “national trauma” (Friedman-Peleg & Bilu, 2011) in a military sense, perpetual security concerns with repeated flare ups tend to make civilian life more stressful and trauma exposure of all kinds more likely (Ben-Ya’acov et al., 2005; Palmieri et al., 2008; Almogy, Kedar & Bala, 2016).

Contrary to our hypothesis, we found no direct link between the frequency of traumatic events and either ED symptoms or BD. We found, however, that emotional dysregulation and dissociation mediated the effect of traumatic events on ED symptoms and BD, and that BMI moderated the effect of traumatic events on ED symptoms and BD. Early studies of trauma and dissociation found that dissociation and past trauma characterized a high percentage of ED patients (Tobin, Molteni & Elin, 1995; Vanderlinden et al., 1993). Later studies focused specifically on dissociative aspects of BN (Hallings-Pott et al., 2005) and BED (La Mela et al., 2010), positing that dissociation may be a way of shifting awareness from negative feelings associated with the trauma towards food and eating. Dissociation may also be used to modulate emotions by narrowing awareness and avoiding thoughts and feelings such as guilt and self-dislike (La Mela et al., 2010).

We found no direct associations between dissociation and emotional dysregulation with BMI. Trauma, ED symptoms and BD were all associated directly with BMI, underscoring the complex web of influences between these variables. Most people tend to under- or overeat when distressed, and emotional difficulties can lead either to high BMI (Czepczor-Bernat et al., 2019) or to low BMI, as in restricted EDs (Farrington et al., 2002) Such alternative patterns make linear associations difficult to find, and path analysis can offer a clearer picture. Furthermore, when participants were divided into high, intermediate and low BMI groups, significant differences were observed only for disordered eating (ED symptoms) and BD, replicating the findings for linear correlations.

A link between trauma exposure and emotional dysregulation has been observed consistently in research and clinical practice (Atchley & Bedford, 2020). Emotional dysregulation has been shown to mediate the effect of childhood trauma on emotional eating (Michopoulos et al., 2015), so that the link between trauma and emotional eating seems partly due to the strong associations between childhood trauma and emotional dysregulation. A systematic review (Palmisano, Innamorati & Vanderlinden, 2016) found that PTSD symptoms, dissociation, depression and stress all mediate the effects of trauma on obesity and BED.

Arguably the most interesting finding of our study was that the frequency of exposure to traumatic events was directly related to BMI, which in turn predicted ED symptoms and BD. Since frequency of traumatic events was not directly associated with ED symptoms, it is our belief that the model may reflect a complex route between trauma and ED symptoms that is not always obvious. These results are supported by a systematic review of traumatic events and obesity, which found a positive association between traumatic events and obesity (Palmisano, Innamorati & Vanderlinden, 2016) and a mediating effect for PTSD and dissociation on the association between traumatic events and obesity. It should be noted, however, that whereas some studies have found that the rate of childhood trauma predicts obesity later in life (Jia et al., 2004), others have failed to find such an association (Goedecke, Forbes & Stein, 2013). It is reasonable to assume that dissociation, emotional dysregulation, BMI and trauma exposure interact in complex ways in a positive feedback loop that accentuates both P TSD symptoms and ED behaviors.

Our moderation analysis attempts to further elucidate these connections. For people with low BMI, those who have experienced few or no traumatic events, ED symptoms were not evident as shown previously (Anderson et al., 2016). However, a different and more complex picture emerged for participants who had low BMI but had experienced many traumatic events. For them, levels of ED symptoms and BD were higher than for those with low BMI and few traumatic. People with high BMI had more ED symptoms and BD, regardless of the frequency of their traumatic events. Although these people tend to have elevated BMI, they may tend to overeat due to trauma-linked emotional distress, rather than ED cognitions and behaviors driven by a drive for thinness. Some high-weight individuals may be eating for numbing, and clinical evaluations of disordered eating and weight problems should therefore include a careful screening for trauma exposure.

This study has several limitations. First, we assessed the frequency of exposure to traumatic events but not their emotional impact or whether they were witnessed rather than directly experienced. A different pattern of results may have been observed had we included a measure of the distress caused by traumatic events. In addition, over half the participants reported that they experience a stressful event not specifically mentioned in LEC, so information about the types of traumatic events experienced is missing. Second, we did not assess post-traumatic symptoms, which may be another trauma-related measure that impacts the study variables. Third, this is a cross-sectional study and cannot lend itself to causational conclusions, Fourth, we operationalized the experience of traumatic life events as their frequency, without taking into consideration the type of traumatic events experienced. Although this approach has been adopted in previous studies of obesity (Palmisano et al., 2018), it is less than ideal. Studies that have examined the effects of specific types of trauma on EDs and BD have generally focused on sexual trauma in childhood, which was not a major focus in this study. Lastly, participants were young Israeli adults, most of whom had been enlisted in the army and grown up in a country where trauma is all too frequent. The results may therefore not be generalizable to other cultures.

This study offers a unique perspective on the influence of traumatic events on ED symptoms and BD. Even though traumatic events may not influence ED symptoms and BD directly, this study suggests that they may affect them indirectly via dissociation, emotion dysregulation and increased BMI. Therapists treating patients with high BMI should be aware of these relationships and investigate the possibility that the sequelae of exposure to trauma may underlie the presenting eating pathology.

Supplemental Information

Supplemental Information 1 SPSS Data concerning all variables in the study.

Click here for additional data file.

Additional Information and Declarations

Competing Interests

Author Contributions

Ethics

Data Availability

Ada H. Zohar is an Academic Editor for PeerJ.

Lilac Lev-ari conceived and designed the experiments, performed the experiments, analyzed the data, prepared figures and/or tables, authored or reviewed drafts of the paper, and approved the final draft.

Ada H. Zohar conceived and designed the experiments, analyzed the data, authored or reviewed drafts of the paper, and approved the final draft.

Rachel Bachner-Melman performed the experiments, analyzed the data, authored or reviewed drafts of the paper, and approved the final draft.

The following information was supplied relating to ethical approvals (i.e., approving body and any reference numbers):

Ruppin Academic Center granted ethical approval to carry out the study within its facilities (Ethical Application Ref: 35 Tashaz).

The following information was supplied regarding data availability:

The SPSS data are available in a Supplemental File.

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
