# Peer review of "Eating for numbing: a community-based study of trauma exposure, emotion dysregulation, dissociation, body dissatisfaction and eating disorder symptoms"

_PeerJ, doi:10.7717/peerj.11899_

## Round 0.1 · original submission · Major Revisions

This manuscript has now been reviewed by two experts in the field. Both reviewers note a number of issues that need to be addressed to consider the manuscript for publication. Having reviewed the manuscript myself, I agree with the points they raised. When revising this manuscript, please respond to each of the reviewer comments, taking note especially of the following major issues:

- Reviewer 1 notes an anomaly in the EDE-Q data, which looks to have been scored on a 1-7 rather than 0-6 scale. Please check whether data were recorded and analysed properly, and ensure these data can be compared to existing research.

- Reviewer 2 notes the lack of theoretical rationale for the proposed hypotheses, and the lack of link between the hypotheses and the literature discussed. I also agree that structuring the narrative to focus on the link between ED symptoms and trauma (as the Abstract is currently structured) would improve the flow of the manuscript.

- Please clarify the rationale for the regression analyses, and clearly describe the results of the SEM and moderation analyses.

Reviewer 1 ·

Basic reporting

Overall the manuscript is well-written, concise, and easy to read. The literature references appear sufficient, given the wide range of topics included in this study (trauma, eating disorders, dissociation, emotion regulation). The raw data is included and shared. The tables and figures are helpful to include. The hypotheses are clearly defined and described within the text.

Please consider the use of person-first language throughout the manuscript; for example, instead of "obese participants" please consider revising to "participants with obesity". Also please add labels to BMI descriptors (e.g., kg/m2). Finally, there is an incomplete sentence listed in line 374.

Experimental design

Please consider updating the manuscript to address the following:

1. How were missing data handled?
2. Some of the EDE-Q data in the SPSS file includes responses such as "7". These are not standard item-level scores for the EDE-Q measure (responses are on a 0-6 scale). Were these scores adjusted in some way? Did the authors intend for these codes to represent missing data? Please be certain these were not accidentally included in total scores for this measure. There are no 0's on this measure for any participant, which seems unlikely. It is very easy to have a 0-6 scale output 1-7 in data software. Yet, this will artificially inflate scores and pathology.
3. The DERS and EDE-Q have multiple subscales. Did the authors just use the total/global score for each of these measures? What scores was included in the analyses? Please clarify within the text to reduce confusion and enhance reproducibility.
4. How was BMI divided into three groups? Which cut-offs were used for each category? Please add additional detail about these analyses.
5. Do the authors have any further information as to the "other" category of traumatic experiences, as this was endorsed by 56.38% of the sample?

Validity of the findings

Once the data has been verified as accurate (see concern regarding the scoring of the EDE-Q measure as noted above), it may be too far of a speculation to ascertain that "eating for numbing" may explain the results of this cross-sectional study. The study did not collect any information about eating patterns or eating for numbing, outside of the EDE-Q which does not directly assess for this behavior. The title of the article implies that the researchers found a relationship with the construct of "eating for numbing" and trauma exposure, emotion dysregulation, dissociation, and body dissatisfaction. Please consider tempering conclusions or editing the title of the manuscript to more accurately represent the study's findings.

Reviewer 2 ·

Basic reporting

1. My main concern pertains to how the rationale for the current study is conveyed. Given that the current study is examining the role of factors in eating disorder symptomatology, I suggest framing the rationale for the current study from the perspective of eating disorder symptomatology (i.e., introductory paragraph should be outlining the problem of elevated eating disorder symptomatology and the following paragraphs describe the potential vulnerability factors).
2. It would be valuable to include any relevant theoretical models that provide basis for the proposed hypotheses.
3. In order to facilitate a more coherent and logical flow of ideas in the Introduction, for each paragraph, I recommend implementing clear opening and concluding sentences.

Experimental design

1. The gaps in the literature are not clearly identified. For example, it needs to be more clearly conveyed that prior research implicates the role of trauma exposure in eating disorder symptomatology, however, it remains uncertain the mechanisms that might underlie this relationship, before then proceeding to describe these mechanisms (i.e., dissociation, emotion dysregulation). I also recommend making clear inferences that then give rise to your hypotheses (e.g., given research showing that X is related to Y and Y is related to B, this gives rise to the possibility that X indirectly predicts B through Y).

Validity of the findings

There are several sections of the results, which need to be clarified. I address each of these below:
1. “To explore the effects of BMI on the various indices, we divided our sample into three groups according to their BMI…” – please clarify the number of people per group. Additionally, it is not clear to me which variables the groups differed on exactly and which variables they did not differ on.
2. Regarding the first hierarchical analysis, please clarify the reasoning behind adding variables at the different steps (e.g., why was BMI added in second step, frequency of exposure to traumatic events & dissociation in the third step, etc.). Furthermore, when describing these results I suggest using the terminology “uniquely associated with” e.g., “As can be seen from Table 4, BMI, dissociation, emotional dysregulation, and BD were all uniquely associated with ED symptoms”
3. Regarding the SEM model, please clarify the indirect associations. That is, you mention that exposure to traumatic events predicted ED symptoms through dissociation, emotional dysregulation, and BMI. However, it is unclear to me whether this was a sequentially mediated relationship or whether these reflect individual relationships (e.g., exposure trauma  dissociation  ED symptoms).
4. Regarding the moderation analyses, as presently described, it is currently unclear to me the observed pattern of results. Given that BMI is the moderator, the pattern of relationships should be clearly described in terms of high/low BMI. For example, the relationship between trauma exposure and BD/ED symptoms was significant only for people with high/low BMI. Please clarify.

Additional comments

Here are some additional minor comments for the authors’ consideration.

- Method, Description of EDE-Q: Please specify whether the measure of ED symptoms was the “global EDE-Q score” and how this was computed (e.g., average of the four subscales).
- Related to one of my previous comments, in the discussion, please clarify the nature of the results which concern the moderating role of BMI, ensuring that these are clearly described in terms of high/low BMI.
- I recommend authors expand on the implications of the rather interesting finding that trauma exposure was found to indirectly associate with ED symptoms e.g., it suggests that trauma exposure leads to greater dissociation, which in turn might lead to greater ED symptoms….
- Additionally, I recommend authors expand on the theoretical and clinical implications of the obtained findings.
- On a final note, another limitation worth noting is that, given the cross-sectional nature of the current study, we are unable to draw conclusions about the causal nature of the observed relationships.

---

## Round 0.2 · Minor Revisions

Both reviewers comment positively on the revised manuscript. Both also request a few remaining minor revisions. Please address each of the following:

1. Upload the data set with corrected EDE-Q scores in the supplementary materials.

2. Refer to global rather than total score for the EDE-Q.

3. Improve the clarity of the final sentence of the opening paragraph

4. For the moderation analysis, conduct follow-up analyses to estimate the conditional effects of the predictor variable on the outcome variable at low and high values of the moderator, including relevant statistics (t, p, and b value).

5. In the discussion, clarify that the SEM results are consisted with a serial mediation model, although further research is needed to confirm such mediation effects.

Reviewer 1 ·

Basic reporting

Please remember to upload the correct data set in the supplementary materials. The only data set I can see through the review portal still has the incorrect scoring for the EDE-Q (1-7 instead of 0-6). It is possible that I may not be able to see the updated data set, in which case please ignore this comment.

Experimental design

Thank you for adding additional information regarding missing data, updates to the statistical analysis section, and clarification around the use of total scores rather than subscale scores for the measure. The EDE-Q global score is now clearly defined. One small note - recommend referring to the EDE-Q score throughout the manuscript and tables/figures as "global" and not "total" score. The use of "global score" is more consistent with the eating disorder literature.

Validity of the findings

The updated theoretical rationale and discussion edits have improved the manuscript. The additional limitations added into the discussion are helpful in clarifying the results from this study.

Additional comments

The authors were quite responsive to feedback. The edits have strengthened the manuscript and provide additional clarity where warranted. No further edits beyond what was mentioned above are requested at this time.

Reviewer 2 ·

Basic reporting

Overall, the authors have done a good job in revising the manuscript and strengthening the rationale for the current study.

I have only one minor suggestion concerning the final sentence of the opening paragraph (i.e., "the current study focuses on..."). Given the importance of this sentence in setting the scene for the current study, I suggest breaking it down into two sentences to improve clarity. e.g.,

The current study seeks to test one potential account of disordered eating hypothesising that exposure to trauma, the absence of effective regulation of negative emotions, and or in the presence of dissociation, contributes to disordered eating behaviours. Such behaviours might serve to numb the unprocessed distress which arise from the trauma the individual has experienced.

Experimental design

No comment.

Validity of the findings

The nature of the results revealed by the moderation analyses are still not clear to me. These analyses seeks to tests whether BMI moderates the relationship between frequency of traumatic events and ED symptoms/BD. Thus, the result should be framed in terms of nature of the relationship between frequency of traumatic events and ED symptoms/BD at low/high levels of BMI (e.g., the relationship between x and y is and positive at high levels of the moderator). To clarify the moderation (interaction) effect, I suggest conducting follow-up analyses to estimate the conditional effects of the predictor variable on the outcome variable at low and high values of the moderator, including relevant statistics (t, p, and b value).

Furthermore, I am concerned that the authors do not fully understand the SEM results, i.e., the statement in the discussion “our model reflects what we believe may be a sequentially mediated relationship”. Please confirm whether the data supports indirect effects.

---

## Round 0.3 · accepted · Accept

Thank you for making these final modifications, in line with the reviewers' requests.